# Organising the Monies of Corporate Financial Crimes via Organisational Structures: Ostensible Legitimacy, Effective Anonymity, and Third-Party Facilitation

**Nicholas Lord** [1,*] 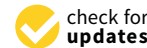**, Karin van Wingerde** [2] **and Liz Campbell** [3]

1   Centre for Criminology and Criminal Justice, University of Manchester, Manchester M13 9PL, UK
2   Erasmus School of Law, Erasmus University Rotterdam, 3000 DR Rotterdam, The Netherlands; vanwingerde@law.eur.nl
3   School of Law, Durham University, Durham DH1 3LE, UK; liz.campbell@durham.ac.uk
*   Correspondence: nicholas.lord@manchester.ac.uk

**Abstract:** This article analyses how the monies generated for, and from, corporate financial crimes are controlled, concealed, and converted through the use of organisational structures in the form of otherwise legitimate corporate entities and arrangements that serve as vehicles for the management of illicit finances. Unlike the illicit markets and associated 'organised crime groups' and 'criminal enterprises' that are the normal focus of money laundering studies, corporate financial crimes involve ostensibly legitimate businesses operating within licit, transnational markets. Within these scenarios, we see corporations as primary offenders, as agents, and as facilitators of the administration of illicit finances. In all cases, organisational structures provide opportunities for managing illicit finances that individuals alone cannot access, but which require some element of third-party collaboration. In this article, we draw on data generated from our Partnership for Conflict, Crime, and Security Research (PaCCS)-funded project on the misuse of corporate structures and entities to manage illicit finances to make a methodological and substantive addition to the literature in this area. We analyse two cases from our research—corporate bribery in international business and corporate tax fraud—before discussing three main findings: (1) the ostensible legitimacy created through abuse of otherwise lawful business arrangements; (2) the effective anonymity and insulation afforded through such misuse; and (3) the necessity for facilitation by third-party professionals operating within a stratified market. The analysis improves our understanding of how and why business offenders misuse what are otherwise legitimate business structures, arrangements, and practices in their criminal enterprise.

**Keywords:** corporate financial crimes; organisational crime; corporate bribery; corporate tax fraud; corporate vehicles; money laundering; illicit finance; proceeds of crime

## 1. Introduction

While it has long been recognised that corporate financial crimes generate financial advantages that substantially exceed those of other serious crimes, such as counterfeiting, illicit drugs, prostitution, and gambling (McGurrin and Friedrichs 2010), research on corporate crimes has mainly focused on their nature and size, their explanations or determinants, their harms and victims, or their regulation and enforcement (for an overview see (Levi and Lord 2017)). The specific issue of how the financial aspects of corporate crimes, in terms of both operational costs and profits generated, are managed remains under-theorised. This is even more apparent as law enforcement authorities in many countries have adopted a 'follow-the-money' approach in the supervision and enforcement of corporate financial crimes (Nelen 2008; Kruisbergen 2017; Kruisbergen et al. 2016) alongside an increasing focus on

the financial organisations (e.g., banks) and professionals (e.g., lawyers, accountants) that enable and facilitate such crimes (Middleton and Levi 2015). In this article, we seek to better understand the intersections of organisations and corporate financial crimes with particular focus on the use of otherwise legitimate organisational structures as a means of controlling illicit finances. Furthermore, we explore the theoretical benefits that can be gained by approaching these issues from an integrated criminological and organisational studies perspective.

The misuse of organisational structures and entities in this way notably came to prominence in the 2016 leak of 11.5 million files at the centre of the Panama Papers scandal, though it has been on the international policy agenda since the start of the 21st Century (OECD 2001). This is not to imply that organisational structures have not been misused historically, but the foregrounding of the issue by the OECD in 2001 was the driver of subsequent policy and scientific agendas. The Panama Papers provided insights into the flows of (illicit) monies through the global financial system and the extensive concealment of legally, unethically, and illegally generated wealth. The spotlight in the case of the Panama Papers fell on Mossack Fonseca, a law firm and company service provider (CSP) based in Panama that specialises in creating offshore companies in jurisdictions such as the British Virgin Islands and the Bahamas to act as conduits for the movement of finances. In 2017, the Paradise Papers leak reaffirmed how such financial arrangements endure transnationally, with the CSP Appleby being scrutinised for its role in facilitating the control of questionable wealth. The implication raised in the Panama and Paradise Papers is that these legal structures are being misused and abused for illicit and illegitimate purposes, such as the evasion and avoidance of tax by wealthy individuals, the concealment of corrupt funds by public officials, and other criminal behaviours, such as money laundering.

With the above in mind, this article analyses how the monies generated for, and from, corporate financial crimes are controlled, concealed, and converted via organisational structures. These organisational structures take many forms, and we focus here on what are termed 'corporate vehicles', which are otherwise legitimate corporate structures and arrangements that facilitate illicit money management. Unlike the illicit markets and associated 'organised crime groups' and 'criminal enterprises' that are the normal focus of money laundering studies, corporate financial crimes involve ostensibly legitimate businesses operating within licit, transnational markets. Within these scenarios, the corporate entity can be the primary offender, an agent of the crime, and/or a facilitator of the management of illicit finances. We focus here on the opportunities presented by these organisational structures in the management of illicit finances that individuals alone cannot access but which require some form of third-party assistance and/or collaboration.

This article is structured as follows. First, we explain what we mean by organisational structures and vehicles, and elaborate on their significance to organisational studies before going on to concretise the intersections of 'corporate crime' and the misuse of otherwise legitimate organisations and organisational structures in corporate financial crimes. Second, we expand on our methodology. Methodologically, there has been no other attempt (that we are aware of) to integrate the identification and assessment of existing empirical materials on the misuse of corporate vehicles using a Rapid Evidence Assessment with insights gained through interview data and case study analysis. Third, we present two case studies from the research—corporate bribery in international business and corporate tax fraud—to ground the nature of the research phenomenon. We use these cases to demonstrate how the findings here are not limited to the idiosyncrasies of specific corporate financial crime types (e.g., corruption and fraud) or specific jurisdictions (e.g., the U.K. and the Netherlands) but have broader global relevance. For instance, corporate financial crimes can differ in terms of their inherent and central processes but all can misuse legitimate organisations to their advantage. Fourth, we analyse three main findings: (1) the ostensible legitimacy created through the misuse of otherwise legitimate business arrangements; (2) the effective anonymity and insulation afforded through such misuse; and (3) the necessary role of third-party professionals that operate within a stratified market. In terms of our substantive contribution to the literature, while the misuse of corporate vehicles has been discussed in the context of organised crime and corruption, it has not been

sufficiently analysed in relation to corporate and white-collar crimes and we begin to address this gap here. Finally, we conclude by arguing that this analysis improves our understanding of how business offenders misuse what are otherwise legitimate business structures, arrangements, and practices in their criminal enterprise.

## 2. Corporate Financial Crimes and Organisational Structures

Conceptually, the focus in this article is on what has been traditionally referred to as 'corporate crime'. That is, those offences, whether criminal, civil, or administrative, that are undertaken by corporate officials (variously dispersed, but representative of the corporate entity) or the corporate entity itself (Clinard and Yeager 1980) or otherwise articulated as offences 'for a firm by the firm or its agents in the conduct of its business' (Hartung 1950, p. 25). That otherwise 'respectable' organisations and corporations are regularly implicated in criminal behaviours is not new; major scandals, such as the LIBOR rigging involving financial institutions including Barclays and UBS, or accounting frauds as with Tesco Plc, or the facilitation of money laundering as with HSBC and Deutsche Bank, give us insight into how the corporation and its environment can be conducive to an array of illicit behaviours for corporate and individual gain at the expense of public and private actors. Empirical evidence has reinforced the widespread, pervasive, and extensive nature of corporate crimes (Sutherland 1983; Clinard and Yeager 1980; Braithwaite 1985; Tombs and Whyte 2015).

It has long been recognised that corporate crimes are also 'organised', both formally and informally (Sutherland 1983, pp. 229–30), and are incentivised and made possible through otherwise legitimate business structures (Levi and Lord 2017). It is this latter point that is of most importance here given our interest in the organisation of the finances of corporate crimes through organisational structures. For instance, it is necessary to understand how corporate offenders confront problems, such as managing the finances for, and from, their criminal behaviours, and the legitimate business structures that shape how, why, and under which conditions they are able to do this over time and place (Edwards and Levi 2008; Lord and Levi 2017). An interesting and important feature of these crimes is that the business offenders have legitimate access to the offending environment (i.e., the organisation and its structures), have spatial separation from likely victims (e.g., market investors), and involve criminal behaviours that appear common and routine within occupational practice providing a superficial appearance of legitimacy and straightforward concealment (Benson and Simpson 2018). Thus, the organisation, or corporation, in addition to providing opportunities and conducive environments for offending behaviours, can be (a) a primary offender, (b) an agent, weapon, conduit, tool, or location for offending, and (c) a facilitator of third party criminality. (Of course, the organisation can also be the victim as well the 'cure' for its own ailments (see Meerts 2018, in this Special Issue)).

The misuse of legitimate organisational structures, and corporate vehicles specifically, has received more academic attention in relation to the concept of 'organised crime' (Ruggiero 2017a) as opposed to corporate and white-collar crimes. For instance, it has been evidenced that organised crime groups may use corporate vehicles to launder illegal profits (e.g., from the drugs trade), to generate income (e.g., boiler room frauds), to avoid personal liability (e.g., as in bankruptcy frauds), or to legitimise other activities (e.g., using a business as a 'front' for illicit market trade) (see Van de Bunt et al. 2007). Additionally, cases of 'blackwashing' or 'reverse money laundering', where legally acquired assets are used to fund criminal activities, have also been analysed (see for example Zabyelina 2015). Money laundering, tax evasion, and bribery in which we see the misuse of 'corporate vehicles' are typically phenomena that transcend the categorical distinctions between corporate and organised crime (see also Ruggiero 2017b). Corporate and organised criminals adopt similar techniques and structures to commit their crimes. At the same time, however, the distinction between the two may inform different institutional responses. Ruggiero (2017a) therefore argues to analyse similarities and differences between the techniques used in corporate and organised crimes. Analysing and comparing the misuse of corporate vehicles in 'organised crime' and 'corporate crime' is a worthy

subject for inquiry, and an issue we explore as part of our Partnership for Conflict, Crime and Security Research (PaCCS) project. However, our focus here is on the opportunities presented by organisational structures in the context of corporate crimes that individuals alone would not be able to realise. These structures enable individuals to manage, conceal, and transfer their illicit finances. This phenomenon, with particular focus on corporate financial crimes, necessitates an analysis of how legitimate business practices and structures facilitate these criminal behaviours by employees internal to and representative of corporations that engage in financial crime in 'glocal', often deterritorialised, markets over time. These organisational structures take many forms (e.g., limited companies, foundations, charities, partnerships) and we focus here specifically on what have been termed 'corporate vehicles' as one organisational form.

## 3. Methodology and Data

The findings are based on data generated as part of a broader comparative project funded by the PaCCS investigating the use of corporate structures in the organisation of serious and organised crimes, including corporate financial crimes. The research is being undertaken in the U.K. and the Netherlands. While the focus is often on offshore financial centres, mainland U.K. and the Netherlands also provide secrecy: the creation of such structures are not the sole prerogative of overseas territories.

We used a mixture of methods to generate data and insights into understanding how, why, and under which conditions those involved in corporate financial crimes misuse corporate vehicles for the concealment, conversion, and control of illicit finance. First, we undertook a Rapid Evidence Assessment (REA) of the available academic literature. Our REA took place between June 2017 and August 2017 and involved an overview of existing scholarship on the topic of 'corporate vehicles and illicit finance'. The purpose of the REA was to develop a 'state of the art' synthesis of the academic literature in the context of an array of 'serious crimes', covering both white-collar and corporate crimes but also behaviours more commonly associated with organised crime, such as money laundering and corruption. Table 1 provides an overview of the REA focus, key primary and alternative concepts, and databases searched. Key words included a mixture of analytical concepts and crime types. Three databases engines were utilised: ProQuest, Scopus, and Web of Science.

**Table 1.** The Rapid Evidence Assessment (REA).

| | |
|---|---|
| **Topic Statement** | The Misuse of Corporate Vehicles in the Organisation of Serious Crimes |
| **Time Period** | Not restricted |
| **Geographical Scope** | Global (English speaking) |
| **Primary Concepts** | 'corporate vehicle', '(offshore) trust', 'limited company', '(offshore) foundation', 'listed company', '(offshore) partnership', 'shell-company', 'shell firm' |
| **Secondary Concepts** | 'illicit finance', 'serious crime', 'white-collar crime', 'crime proceeds/proceeds of crime', 'dirty money', 'money laundering', 'fraud', 'criminal enterprise', 'organised crime', 'corporate crime', 'financial crime', 'tax evasion', 'offshore' |
| **Databases searched** | - ProQuest (44 Databases: see website) <br> - SCOPUS <br> - Web of Science |

*Primary* and *secondary inclusion criteria* were then applied to the search query hits. Table 2 provides an overview of the search queries and hits in addition to the sources that met the inclusion criteria. The search included only peer-reviewed journal articles that had 'full text' availability and that were written in English. There was no date range restriction. All hits were then manually examined to determine their relevance to the topic statement. As a consequence, many hits from the search queries were omitted. This resulted in a total of 132 relevant academic publications following the application of the primary inclusions criteria. The application of the secondary criteria involved a further manual

sift of the hits to identify only those publications based on empirical research and of direct relevance to the REA question. These stringent criteria narrowed the number of relevant hits substantially to 21 articles with an empirical underpinning of relevance for our research question.

**Table 2.** REA Search Hits Following Primary and Secondary Inclusion Criteria.

| Primary Concept | Query | Secondary Concepts | Primary Inclusion Criteria | Secondary Inclusion Criteria |
|---|---|---|---|---|
| "corporate vehicle" | | | 28 | 7 |
| "trust" | | ("illicit finance" OR "serious crime" OR "white-collar crime" OR "white collar crime" OR "crime proceeds" OR "proceeds of crime" OR "dirty money" OR "money laundering" OR "fraud" OR "criminal enterprise" OR "organised crime" OR "organized crime" OR "corporate crime" OR "financial crime" OR "tax evasion" OR "offshore") | N/A | N/A |
| "offshore trust" | | | 13 | 1 |
| "limited company" | | | 14 | 3 |
| "foundation" | AND | | N/A | N/A |
| "offshore foundation" | | | 0 | 0 |
| "listed company" | | | 20 | 1 |
| "partnership" | | | N/A | N/A |
| "offshore partnership" | | | 4 | 0 |
| "shell company" | | | 47 | 9 |
| "shell firm" | | | 6 | 0 |
| | | | 132 | 21 |

N/A = not applicable.

The texts of all publications were imported into NVivo.[1] All texts were then read by each of the investigators and analysed in terms of their methodological quality and rigour and the relevance of the empirical findings for answering our research question.

Second, semi-structured interviews were conducted with 35 actors from law enforcement, public authorities, financial institutions, non/inter-governmental organisations, professional services, and academia primarily in the Netherlands and the U.K. This included an expert group meeting with 11 key actors from enforcement authorities, professional services (law firms), and academia in July 2017. The interviews and expert group meeting were designed to understand when and why the use of corporate vehicles might be problematic, the definitional and legal landscape surrounding corporate vehicles, the nature and organisation of the misuse of corporate vehicles for financial gain, and regulation and enforcement of the misuse of corporate vehicles and possible obstacles inhibiting successful enforcement. All interviews lasted on average an hour and were thereafter transcribed. All interviews were analysed using NVivo. The interviews were analysed iteratively, meaning that there were constant shifts between the data and the literature and regular meetings were held to discuss and interpret the data.

Our analysis below (see 'Discussion') is directly informed by the literature identified in our REA and our interview data. We arrange our discussion around three prominent themes that emerged during the analysis of the literature and during our interviews, and draw on these sources to build our conceptual and theoretical insights into the nature and purpose of misuse. In this sense, we triangulate our data to corroborate our core findings. We do not include all literature identified as part of the REA as some was not directly relevant to the organisational aspects of corporate crimes but was concerned with 'organised crime'. Our analysis does not include direct quotes from our interviews but is arranged to discuss core themes in a more integrated, narrative style.

---

[1] NVivo is analysis software designed for managing qualitative data but can also be utilised in the coding of literature.

### 4. Case Studies

Corporate financial crimes take many forms. In order to concretise the nature of how organisational structures can be misused, we present two cases from our analysis concerned with corporate bribery in international business (i.e., corruption in commerce) and corporate tax fraud (i.e., intentional dishonesty at the expense of public funds). These cases were selected to provide an illustrative account of *how* organisational structures can be misused in corporate financial crimes. In these terms, the cases are not necessarily representative of all cases we encountered but are useful as heuristics to stimulate further investigation. First, we discuss the BAE Systems bribery case. We used open sources in our description and analysis of the case; specifically, court documents and media reports. The second case was derived from the Dutch Organised Crime Monitor, an ongoing systematic analysis of closed large-scale police investigation into organised crime in the Netherlands. It has existed since 1996 and aims to gain insight into the nature of organised crime in the Netherlands and its developments and to use this knowledge to optimise the prevention and fight against organised crime.[2] We chose a case of corporate tax fraud that illustrates how these corporate structures are being misused.

#### 4.1. Corporate Bribery in International Business

When organisations, and corporations in particular, are implicated in bribery in international business, it means those actors operating within the organisation (i.e., employees or senior managers), or on behalf of the organisation (e.g., intermediaries, subsidiaries, or agents), have engaged in an illicit relation of exchange with a foreign public official (or their agent), either as instigator or on request, to win or maintain a business advantage for their organisation (Lord 2014a; Lord and Doig 2014). Bribery in business is a core focus of international conventions such as the OECD Anti-Bribery Convention 1997 and the UN Convention against Corruption, with such behaviours now constructed as universal social bads, particularly in those countries with large shares of world exports (i.e., key players in international commerce, though some remain absent, such as China and India) (Lord 2014b, 2015). Consequently, enforcement and regulation domestically is a priority concern for nation states seeking to communicate an image of active enforcement of such criminality (and perhaps normative superiority) to those international moral entrepreneurs, such as Transparency International, that scrutinise how they respond to their corporations that bribe. The misuse of 'corporate vehicles' is common in, and at the centre of, the organisation of many cases of corporate bribery and related finances.

The Case of BAE Systems

In 2010, BAE Systems (BAES), the U.K.'s largest arms manufacturer, agreed to pay criminal fines in the U.S.[3] ($400 million) and U.K.[4] (£0.5 million) to settle charges related to failures in accounting and bookkeeping but in connection to allegations of bribing foreign public officials in Saudi Arabia, Tanzania, the Czech Republic, and Hungary to win or maintain arms contracts.

In the case of Saudi Arabia, bribes totalling over £6 billion were allegedly paid to Saudi officials as part of a series of defence contracts signed between the U.K. and Saudi Arabian governments. In 1985, the initial al-Yamamah I arms deal (al-Yamamah II was signed in 1988) involved the provision of defence equipment, such as Tornado and Hawk aircraft, in exchange for up to 600,000 barrels of oil a day. The deal was worth around £43 billion. However, finances for the bribes were created by inflating prices to enable 'kickbacks' to be paid which covered extravagant expenses, such as yachts, sports cars, a private jet, and cash payments. The finances for these inducements were organised through shell

---

companies located in offshore locations and bank accounts in more secretive jurisdictions. For instance, according to court documents, BAES made a series of substantial payments to shell companies and third-party intermediaries that were not sufficiently scrutinised but were used to conceal the use of 'marketing advisors' and the provision of 'support services':

[2]   The second author had access to these cases through a previous commissioned study in anticipation of legislative changes in the Netherlands with regard to the supervision and control of corporate vehicles (Van de Bunt et al. 2007).
[3]   https://www.justice.gov/opa/pr/bae-systems-plc-pleads-guilty-and-ordered-pay-400-million-criminal-fine.
[4]   https://www.caat.org.uk/resources/companies/bae-systems/r-v-bae-sentencing-remarks.pdf.

> BAES contracted with and paid certain advisors through various offshore shell companies beneficially owned by BAES. BAES also encouraged certain advisors to establish their own offshore shell companies to receive payments from BAES while disguising the origins and recipients of these payments. (DoJ 2010, para. 8)

Figure 1 provides a visualisation of these organisational arrangements. In one such instance, BAES established a shell company called *Red Diamond Trading International Ltd.* in the British Virgin Islands (BVI) in order to: (i) conceal its marketing advisor relationships (identity and payments); (ii) create obstacles for investigating authorities; (iii) to circumvent laws prohibiting such relationships; and (iv) to assist advisors in avoiding tax liabilities for payments from BAES. Through *Red Diamond*, BAES made payments of more than £135 million despite being aware the funds would be used to influence contract decisions in foreign governments. In another instance, in one 20-month period, BAES paid over £8 million to a front company called *Robert Lee International (RLI)* created by BAES to entertain top Saudi officials[5] with payments transferred via intermediary-owned bank accounts in Switzerland. Payments were also concealed through other front companies created by BAES.[6]

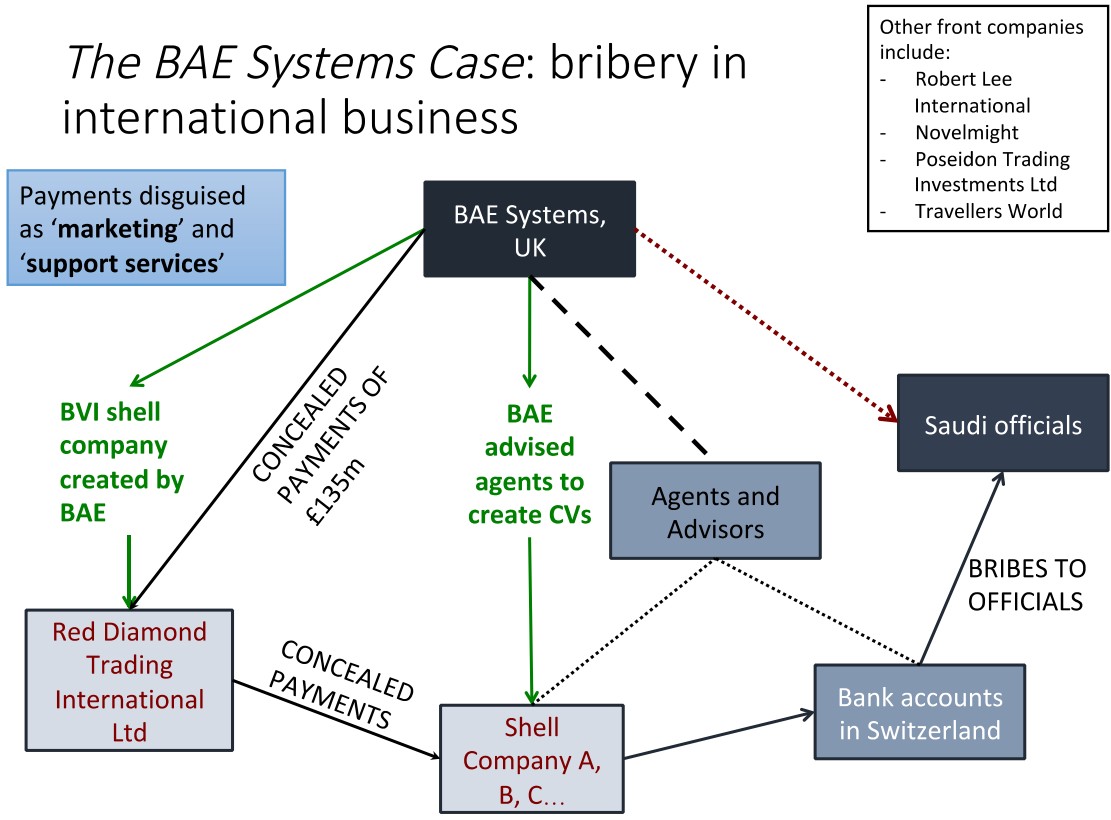

**Figure 1.** Corporate vehicles (CVs) and BAE Systems Bribery Scandal.

---

5    https://www.theguardian.com/world/2003/sep/11/bae.freedomofinformation.
6    https://www.theguardian.com/baefiles/page/0,,2095840,00.html.

The use of organisations as structures for illicit finance in this way in cases of corruption is common. An analysis of 213 'grand' corruption cases between 1980 and 2010 identified that over 70% (150) involved the use of at least one corporate vehicle that concealed, at least in part, beneficial ownership. In total, 817 corporate vehicles were used in those 150 cases and the U.K. and its crown dependencies and overseas territories had the second highest number of registered corporate vehicles behind the U.S. (Van der Does de Willebois et al. 2011).

*4.2. Corporate Tax Fraud*

Tax fraud covers a range of behaviours that involve deception and/or dishonesty for financial gain, such as tax evasion and other forms of non- or under-payment of tax liabilities. A broader conceptualisation that is not limited by criminal law doctrine might focus on tax non-compliance, where we see tax avoidance and aggressive tax planning, particularly by large multi-national corporations. These behaviours are characterised by an improper transfer of money to those evading tax and away from public funds (Leighton 2010, p. 526). Tax frauds reduce tax performance that in turn can increase the tax burden for those who are compliant (Torgler 2010, p. 535). The creation of offshore corporate vehicles through which to conceal, convert, and control finances generated through tax fraud is a common modus operandi. Offshore secrecy havens and the use of corporate vehicles enable rich individuals and corporate elites to pay small amounts of tax and facilitate the concealment of tax fraud schemes and associated proceeds (Levi 2010, p. 495).

The Case of Jansen BV

This case illustrates the combination of tax fraud and the laundering of its proceeds using various corporate vehicles. This case revolves around the Dutch textile wholesale company Jansen BV[7] that had been importing textiles from China and Hong Kong for many years. Mr. Jansen was the CEO of the company and its only shareholder. In order to pay less tax, the following simple but effective construction was set up. First, Mr. Jansen bought the shares of two corporate vehicles abroad. The first vehicle is Wemax Ltd., an offshore company established and based in Hong Kong. The other company is Tejeko NV, which was established in the Dutch Antilles. Mr. Jansen was the sole shareholder of both companies and has full control over both companies. The management of these companies was, however, based in local trust offices in Hong Kong and Curacao.

Wemax Ltd. was placed between Jansen BV and the supplier of textiles in Hong Kong. On paper, Wemax Ltd. purchased the textiles and then sold them to Jansen BV who paid its dues on Wemax's foreign bank account. Consequently, the original supplier of the textiles in Hong Kong had now been concealed for the Netherlands Tax and Customs Administration and it had been made to look as if Jansen BV only did business with Wemax Ltd. Wemax Ltd. then fictitiously doubled the price for the textiles: the original purchase price of €750,000 was raised on paper to €1,500,000. The invoices were addressed to Jansen BV that paid these and included the invoices in its annual reports and tax returns. Wemax Ltd. thus received €1,500,000. Of this, €750,000 was paid to the original supplier of the textiles and the remaining €750,000 was actually 'saved' on the bank account of Wemax Ltd. On paper, the origin of the money is legitimate—from the sale of textiles—and could be withdrawn from the company without actual taxes being paid. After a few years, Jansen had saved approximately €3,000,000 on Wemax Ltd.'s bank account in Hong Kong.

In order to actually use this money, Jansen needed the second corporate vehicle, Tejeko NV, based in the Dutch Antilles. Jansen wanted to use this money to buy a new office building in the Netherlands. Jansen's financial advisor recommended him to take out a loan from Tejeko NV for the amount of €3,000,000. As the money was still in the bank account of Wemax Ltd. in Hong Kong, the advisor also suggested to take out the money in cash as a bank transfer is easily traceable for the authorities.

---

7    Names are fictitious.

Therefore, Jansen flew to Hong Kong repeatedly and took out €3,000,000 in cash from Wemax Ltd.'s bank account. After the cash had been deposited on Tejeko NV's account, Tejeko NV provided a loan of €3,000,000 to Jansen BV in order to purchase the office building. The loan then appeared as a debt on Jansen BV's balance sheet. The following schematic illustrates this structure (Figure 2):

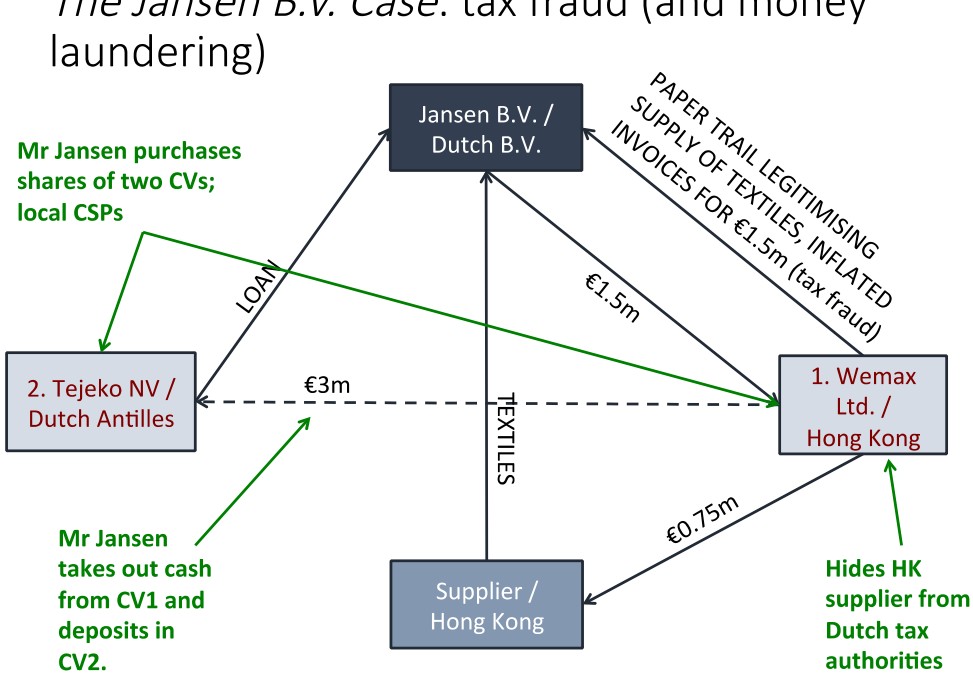

**Figure 2.** CVs and the Jansen BV tax fraud.

Through this scheme, Jansen has washed the money that he obtained criminally by tax fraud. After all, on paper, the money has been given a legitimate origin, namely a loan from a corporation in the Dutch Antilles. At the same time, he concealed the fact that he actually lent his own money. The result of this scheme is that the profit of Jansen BV was reduced artificially, which means that he had to pay less tax per year. Moreover, Jansen was withdrawing money from his company with which he created a pot of black money abroad that he could freely use afterwards. Also, in this case, the corporate vehicles have been specifically created or purchased for this purpose and their characteristics are attractive for misuse. The creation of the vehicles in Hong Kong and the Dutch Antilles does not require any minimum capital, which makes the setup relatively cheap. Offshore companies from Hong Kong are further characterised by the ability to guarantee anonymity of shareholders. Finally, this combination of vehicles in different jurisdictions, especially Hong Kong, creates problems for law enforcement authorities. Requesting information from these jurisdictions is extremely time-consuming and often fails.

## 5. Discussion

In this section, we integrate varied data sources to discuss three key findings in the organisation of illicit finances through organisational structures and *corporate* vehicles specifically. In particular, we focus here on (1) the ostensible legitimacy created through the misuse of otherwise legitimate business arrangements and practices in the use of organisational structures; (2) the anonymity and insulation afforded through such misuse; and (3) the necessary role of third-party professionals as (witting/unwitting) facilitators that operate within a stratified market. These three features are interconnected and overlap, as ostensible legitimacy and virtual anonymity cannot be accessed without third party assistance.

*5.1. Ostensible Legitimacy: The Misuse of Legitimate Business Arrangements*

The first key finding is that the misuse of certain organisational structures provides a veneer of legitimacy. This allows actors involved in corporate financial crimes to give their illicit behaviours a superficial appearance of legitimate action given the close proximity of these financial arrangements to normal business practice. Thus, the illicit practices are concealed. We use the term 'ostensibly' to reflect that while the structures misused are, technically, both legitimate and lawful, in the context of criminal misuse they have only an illegitimate purpose. (Though they could also be used for legitimate purposes). This is important to recognise as it is the ostensible nature provided, rather than the criminal misuse of something legitimate, that is the attractive feature.

According to the OECD (2001, p. 13), corporate vehicles can be defined as 'legal entities through which a wide variety of commercial activities are conducted and assets are held'. These vehicles include a range of organisational forms, often referred to as shell companies, and often have limited liability features. Furthermore, these legal structures permit businesses to incorporate companies in low- or no-tax regimes, provide flexibility in global markets, and reduce the level of regulation, particularly when set up in offshore financial centres that offer great secrecy, either by concealing the origin of the money or the identity of—what has become known as—the ultimate beneficial owner.[8] In these terms, the use of sophisticated corporate vehicles and structures 'to hide the origins of investments or to conceal beneficial ownership of property are legitimate' (Nelen 2008, p. 755). Thus, large flows of money, wealth, and assets move through or are controlled via the global financial system in this way, with (offshore and onshore) financial centres and companies enabling rich global elites, both individuals and companies, to manage their finances for varied legal (and illegal) purposes.

However, while their creation could be primarily for licit purposes, '[t]ransactions processed through the corporate account of a "shell company" become effectively untraceable and thus very useful for those looking to hide criminal profits, pay or receive bribes, finance terrorists, or escape tax obligations' (Sharman 2010a, p. 129). Since 2001, a number of intergovernmental and nongovernmental organisations have highlighted concerns over corporate vehicles being used to conceal criminal monies (OECD 2001; FATF/OECD 2006; Van der Does de Willebois et al. 2011; Otusanya and Lauwo 2012; Global Witness/Christian Aid 2012; Transparency International 2014). Thus, while corporate vehicles predominantly are used for legitimate purposes (e.g., transnational commerce and associated practices, such as mergers and acquisitions or tax planning), they also present opportunities for those involved in criminal enterprise to conceal and control illicit funds whilst maintaining anonymity through the obscuring of 'beneficial ownership' (FATF/OECD 2006, p. 1). Corporate financial crimes, such as bribery and tax frauds, generally involve finances that are already embedded within legitimate financial arrangements (e.g., contractual relations) and contexts (e.g., in bank accounts). This necessitates the use of other organisational structures to transfer the money, as moving such large amounts is improbable via cash or other value systems, although a company could provide credit cards or cash cards for use by perpetrators or beneficiaries both domestically and internationally.

That business offenders portray a pretence of respectability to disguise their underlying deviant behaviour has long been recognised in the criminological literature (Ross 1907; Sutherland 1983). This inherent duplicity and superficial appearance of legitimacy has emerged as a key feature in how most white-collar and corporate criminals realise opportunities for crime and remain undetected (Benson and Simpson 2018). For instance, the use of corporate vehicles as conduits for finance in itself

---

8　　The Fourth EU Anti-Money Laundering Directive (2015) defines beneficial ownership as 'any natural person(s) who ultimately owns or controls the customer and/or the natural person(s) on whose behalf a transaction or activity is being conducted' (see also FATF/OECD 2016 and FATF/OECD and CFATF 2010). Thus, a 'beneficial owner' is 'a natural person—that is, a real, live human being, not another company or trust—who directly or indirectly exercises substantial control over the company or receives substantial economic benefits from the company' (Global Witness 2013, p. 3). Thus, key features are the control exercised and the benefit derived by those people that own the company (Van der Does de Willebois et al. 2011, p. 3).

does not indicate misuse. Similarly, in normal economic trade, it is perfectly legitimate not to act in one's own name but to take on another 'identity'. Business offenders are able to misuse these otherwise genuine financial arrangements as they construct ostensibly legitimate arrangements to obscure their underlying criminality. Organisational cultures can be conducive to those looking to rationalise such behaviours, making generating pro-social and ethical climates within corporations essential to reduce potential criminal behaviours (see Gorsira et al. 2018, in this Special Issue).

Licit corporate entities provide opportunities to act as structures for the concealment, conversion, and control of illicit finance by offering an external appearance of legitimacy to the 'beneficial owners' of these entities and/or the clients who use them to transfer funds. The hiding of true beneficial ownership in this way has been identified as the most significant feature of the misuse of corporate vehicles (FATF/OECD 2006, p. 2), and this is borne out in our interviews. This ostensible legitimacy is constructed in three primary ways:

- *Organisational forms*: business offenders are able to set up various organisational forms (e.g., limited companies, shell corporations, etc.), usually with assistance from third-party specialists (see below), to construct an ostensibly legitimate ownership arrangement. For instance, individual 'A' can be the beneficial owner of companies 'X', 'Y', and 'Z' and use these to conceal their own involvement.
- *Organisational relations*: business offenders can construct fabricated trading relations between those structures that have been arranged. For instance, individual 'A' can enter companies 'X', 'Y', and 'Z' into contractual or service arrangements that have no substance but enable falsified records to be generated.
- *Organisational practices*: once structures and relations are in place, business offenders can generate fictitious invoices and paper trails to enable finances to transfer via these structures (or appear to) in order for the true underling illicit monies to be concealed and legitimised. For instance, company 'X' can send electronic invoices to company 'Z' that acts as an interlocutor to company 'Y'. This layering approach can further obscure beneficial ownership and illicit finance.

Key to most cases of misuse is that the relationship between the natural persons (i.e., the ultimate beneficial owners) and the corporate vehicles is either concealed or proposed differently than in reality. In most cases of misuse, shell companies are used to conceal or convert the finances or the identity of the ultimate beneficial owners. Their life cycle strongly depends on the fictitious role that they play in the structure. Central to this fiction are practices of falsification, such as through fake invoices (i.e., inflated prices to reduce tax liabilities) or fabricated services (i.e., 'marketing' and 'support services' to disguise monies for bribery). In these cases, we see the parasitical nature of such occupational and organisational deviance as the business offenders implicated hide their criminality behind practices that have an appearance of legitimacy (Benson and Simpson 2018). This creates obstacles to detection for enforcement authorities.

## 5.2. Effective Anonymity (and Insulation)

The second key finding is that of effective anonymity. This allows those individuals involved in corporate financial crimes to conceal their identity, and offset possible intervention or enforcement of the law, providing a layer of insulation. Thus, the illicit actors themselves are *effectively*, but not entirely, concealed as there will always be some level of connection between the actors and the finances even where this is well-obscured. Corporate vehicles are attractive for criminals and unethical individuals and groups as they are set up in secretive jurisdictions that provide anonymity to their owners and the transactions processed through them effectively become untraceable. Thus, anonymity is a central purpose of using a corporate vehicle. In questioning why request anonymity, rather than a complex legal trail, Sharman (2010a, p. 133) states that '[e]ven if the legal trail is complex, as long as the service provider has proof of the identity of the ultimate beneficiary of a firm, the veil of secrecy is vulnerable to being pierced [despite it being difficult, time-consuming, and expensive to investigate]

. . . However, if no information is collected by the service provider in the first place, nothing can be disclosed later'. With this effective anonymity in mind, as Findley et al. (2013, p. 658) note, '[s]hell companies that cannot be traced back to their real owners are the standard vehicle of choice for those looking to hide illicit financial flows'. By using shell corporations as nominal account holders, extra layers of secrecy between bank accounts and beneficial owners can be created, essentially making such accounts equivalent to numbered accounts that now are prohibited by anti-money laundering regulations (Johannesen and Zucman 2014, p. 85).

The creation and misuse of corporate vehicles is often associated with so-called offshore financial centres and offshore tax havens. However, this may not necessarily be the reality. Sharman states that, in contradiction to conventional wisdom, his findings 'cast strong doubt on the proposition that the problem of financial opacity is caused by palm-fringed tropical islands, rather than large higher-income economies like the United States and Britain' (Sharman 2010a, p. 134). Indeed, our interviewees emphasised the misuse of companies and Scottish limited partnerships in the U.K. to maintain illicit assets (Campbell 2018a). While nearly all offshore centres regulate CSPs, the U.S. and Britain have chosen not to. Thus, 'powerful states are choosing to profit by not following the standards they have imposed on others' (Sharman 2011, p. 984). More generally though, the use of financial centres or tax havens, whether offshore or onshore, represents the pursuit of a 'calculated ambiguity' (Sharman 2010b, p. 2) as it permits obscurity to those looking to conceal (illicit) wealth. They also permit actors involved in criminal behaviour to separate themselves jurisdictionally from the victims of their crimes and from the enforcement and regulatory authorities, creating insulation to offenders.

Making transparent the true beneficial owner anonymised through offshore corporate structures has been identified as central to responding to 'a range of high-priority international problems: the drug trade, organised crime, terrorism, money laundering, tax evasion, corruption, corporate crime, and systemic financial instability' (Sharman 2010a, p. 129) and is at the core of the E.U.'s Fourth Anti-Money Laundering Directive (Campbell 2018b).

In Pursuit of Increased Transparency

Given concerns over the identification of the ultimate beneficial owners and the anonymity they can permit, companies are now required to register any owners with at least 25% stake in the company, and this may well be reduced to 10% in the future (Campbell 2018a, 2018b). However, arbitrary ownership thresholds are straightforward to circumvent as ownership can be split to fall within the threshold limit, ensuring continued anonymity. In other words, actors are able to structure ownership creatively in order to evade being registered as 'beneficial owners'. Similarly, data analysed from the first submission to the U.K.'s Public Register indicates that many requirements are not being met. Analysis by Global Witness indicated that '[a]lmost 3000 companies listed their beneficial owner as a company with a tax haven address, something that is not allowed under the rules', amongst other concerns about data inputting.[9] Thus, such registers may be 'utopian' and ineffective in the current environment given it depends on if and how they will be monitored and enforced. The key question therefore is who polices and monitors these registers and how inaccurate or opaque materials can be challenged. Enforcement of the rules is fundamental to pursuing transparency. This is further corroborated by the U.K.'s public authorities we interviewed, which pointed that out there is a 'compliance cost' to investigating misuse. For instance, obstacles are created as corporate vehicles may be set up in the U.K. but trade overseas. A further major stumbling block remains over the transparency of companies registered in overseas U.K. territories although in 2018 the U.K. government committed to ensuring these territories implement public registers by the end of 2020.

---

[9]　https://www.globalwitness.org/en/blog/what-does-uk-beneficial-ownership-data-show-us/.

*5.3. Third-Party Facilitation and Market Stratification*

Our third finding is that the ostensible legitimacy and effective anonymity provided to individuals using corporate vehicles as structures for illicit finance almost always requires the involvement and facilitation of expert or professional third-party collaborators. Corporate vehicles can be relatively straightforwardly created and/or dissolved in onshore and offshore locations without proof of identity followed by establishing bank accounts for these entities (Sharman 2010a; Van de Bunt et al. 2007). Ownership structures can take many forms, with shares being issued to natural or legal persons in registered or bearer form, and they can have single or multiple purposes (FATF/OECD 2006; OECD 2001). Our research indicates that third-party legitimate actors (e.g., accountants, lawyers, other professionals) are necessary relations in the organisation of illicit finances in cases of corporate financial crimes. These actors can become witting, or unwitting (or wilfully blind), facilitators in particular when acting as CSPs to set up, service, and sell corporate vehicles and other shell companies (Ruggiero 2017a; Chaikin and Sharman 2009, p. 75; FATF/OECD and CFATF 2010; Lankhorst and Nelen 2005; OECD 2001). Such CSPs enable those engaged in illicit market and commercial enterprise to engage in legitimate business transactions and relations and to obscure the provenance and ownership of income, wealth, and assets by exploiting legal/regulatory lacunae and differences in legal/enforcement regimes.

A central issue for criminal actors is that in order to conceal their finance, some form of collusion and/or cooperation with external, professional actors, such as accountants, lawyers, and other professionals, may be required, though informational shielding and distortion may reduce the risks for them. As Levi (2015, p. 10) notes, 'this involves trust in a particular person or persons—perhaps a member of one's close or extended family or ethnic/religious group—or trust in an institution, such as a bank or a money service business (MSB) or a lawyer who may be a trustee of a corporate entity, to an extent sufficient to defeat whatever level of scrutiny will actually be applied'. A key question to ask therefore is how are these third-party actors recruited and how much do CSPs and facilitators actually know (or should do all they can to know) about the misuse of the companies that they create and are they complicit in their misuse for illicit purposes?

Empirical research has identified that 'more than one in four providers (26.2%) worldwide is willing to violate international standards by offering incorporation without certified proof of customer identity, meaning that in practice anonymous shell companies are readily available' (Findley et al. 2013, p. 660). Consequently, Findley et al. (2013) conclude that 'there is a substantial level of noncompliance with the international standards mandating that providers obtain certified identification documents from beneficial owners when forming shell companies' (p. 681) and that 'service providers are no more likely to comply with international rules when they are prompted about the existence and content of the rules' (p. 681). Similarly, in Sharman (2010a) audit study of compliance with the prohibition of anonymous shell companies, 45 providers responded to offer their services, of which 17 offered to set up an anonymous vehicle. Thirteen of the 17 successful approaches were to company service providers in OECD countries compared with only 4 in 28 countries labelled as 'tax havens'. In both studies, a key limitation was that they sought to enlist CSPs available publically but it can be expected that those global corporate and individual elites access CSPs via social network, making insights into these hidden connections difficult.

In these terms, our research indicates that a stratified market exists whereby CSPs that offer services publically online form only one segment. Other, more esoteric, CSPs do not advertise but rather involve introductions through personal networks or relations established at high-level events for elite clientele. Gaining access to these hidden service providers is problematic. In policy terms, increased attention to and responsibility of third-party intermediaries and actors who, whether knowingly, with willful blindness or through incompetence, facilitate the concealment of illegal behaviours is needed. Increased oversight of and intervention with these third party legal and accounting professionals is vital, not least as some seem to operate with impunity.

The Specific Role of Banks and Financial Institutions

Banks and financial institutions have a necessary role in enabling these illicit arrangements and as Ruggiero (2017a) notices offer a number of disconnects between the act committed and the beneficiaries of the crime. First, for illicit finances to be transferred via corporate vehicles, bank accounts need to be established. Second, banks also provide the infrastructure to allow monies to be transferred across the accounts of individual entities and for transactions to take place. Thus, banks and financial institutions are essentially the entry points to the financial system. Without banks and the financial system, the management of illicit finance cannot function. Given this central role, they are required to implement strict anti-money laundering requirements in relation to the 'onboarding' of new corporate and individual clients and the monitoring of suspicious transactions.

In terms of 'onboarding', our research has indicated that corresponding due diligence and client checks can be time-consuming and pervasive, in some cases up to eight months because of extensive checks, which in turn create pressures for financial institutions. Banks face internal and economic pressures not to lose clients and this is exacerbated when clients threaten to move to other providers where difficult questions are in some way circumvented. For instance, one concern for established financial institutions is the emergence of 'Challenger Banks' that promise clients to undertake due diligence processes much quicker but in doing so allow for more risk. While it is unclear whether or not there will be less oversight and more blind spots within challenger banks, it can be expected that it will be difficult for challenger banks to find the balance. However, as they tend to be innovative, they may find solutions to the 'problem'. As a consequence, there are cases whereby financial institutions, such as banks, are 'far too willing to do business with anonymous companies' (Global Witness 2013, p. 7). Banks are required to implement sufficient internal systems to identify and then report suspicious transactions and money laundering, but risk-based approaches to anti-money laundering are not consistent across the banking sector. Furthermore, 'the identification of "suspiciousness" by professionals and others with a legal responsibility to combat money laundering is often a judgment that the people and/or transactions are "out of place" for the sort of account they have and the people they purport to be' (Levi 2015, p. 10). We must question how, in cases such as BAES and Jansen BV, those involved are able to conceal their illicit behaviours from others.

## 6. Conclusions

Our core argument in this article is that organisational structures, and corporate vehicles in particular, provide opportunities to individuals involved in corporate financial crimes to enjoy and be insulated by an ostensible legitimacy and effective anonymity for the criminal behaviours and the criminal actors, respectively. Without the use of these organisational structures, individual actors would not be able to access such concealment opportunities in the course of their criminal behaviours.

For instance, corporate vehicles can be used to launder illegal profits. In the case of Jansen BV, a structure was created whereby Jansen retained anonymous control over the money generated through the tax fraud. The structure was aimed at misuse, since the corporate veil legitimised investing money from tax fraud into legitimate assets. However, corporate vehicles can be used to legitimise other activities. This is not primarily aimed at financial gain and includes situations in which the natural person does not hide behind the corporate veil, but rather uses it to disguise illegitimate activities. For example, in the case of BAES we see how corporate vehicles are created to provide an appearance of transactional legitimacy in the concealment of finances used for bribery. In both scenarios, these arrangements have been used to avoid personal liability. In the BAES case, for example, agents and advisors hid behind corporate vehicles constructed for illicit purposes with any commissions attached to bank accounts in the name of the corporate entity but under the control of those individual actors.

Of course, organisational structures can facilitate corporate crimes in more ways than organising finance. For example, in his research on the Madoff scandal, Van de Bunt (2010) shows that Madoff used the organisational structure of his firm to create isolation and to conceal the fraud from non-complicit employees. Madoff's Ponzi scheme originated from a separate department which was

located on a separate floor. Thus, large and complex organisations may provide cover for corporate crimes through the division of tasks, decentralisation of decision-making, and specialisation of work. This stresses the importance of research into how corporate crimes are organised and facilitated through complex business structures. These arrangements are able to endure and withstand intervention as enforcement asymmetries, obstacles to cross-border information exchange, and cultures of corporate non-compliance globally create barriers to regulatory responses (Sharman 2010a, p. 138). For instance, our project's Expert Group Meeting attended by informed actors from public authorities, amongst other key organisations, identified tensions between (political and economic) openness to foreign investments/investors and the prevention of crime. Governments must seek to protect national economic interests whilst also communicating an image of enforcing strict international standards. Furthermore, the issue of information exchange and actually receiving relevant information from offshore jurisdictions was highlighted as a primary obstacle, demonstrating notable imbalances in enforcement structures and capabilities across jurisdictions.

By integrating criminological insights into the dynamics of the financial aspects of corporate financial crimes with an appreciation of the significance of the study of organisations and their features and associated practices, we have been able to gain theoretical insights into how and why the organisational form can provide opportunities for crimes that individuals alone cannot access. However, we recognise the need for further empirical research in this area to illuminate the connections between the location of individual and corporate criminality within an organisation and their place within more enduring financial arrangements and systems.

**Author Contributions:** All authors contributed to the design of the research, data collection and analysis and co-wrote the article. All authors read and approved the final manuscript.

**Funding:** This research was funded by the Partnership for Conflict, Crime and Security Research (PaCCS) grant number [ES/P001386/1].

**Conflicts of Interest:** The authors declare no conflict of interest.

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
