# Peer review of "Organising the Monies of Corporate Financial Crimes via Organisational Structures: Ostensible Legitimacy, Effective Anonymity, and Third-Party Facilitation"

_admsci, doi:10.3390/admsci8020017_

Round 1
Reviewer 1 Report
This is a well written article and one important advantage is that it is very clear how companies are used for financial crime. It is very pedagogical in that respect. I don´t think that very musch is new, but the two case studies are interesting to read.
One important explanation to use Corporations for financial crime is that the Money is in an account Environment. In order to transfer a lot of Money from an account you need other companies.
Even if counterfeiting is not a typical financial crime, it is often a Corporate crime.
Organised crime often use companies these day´s. One explanation is the increase of fraud and that you need an account to get the Money from fraud. This is an effect of the cash free society and that the administrative systems have been more complex, which one will misuse for fraud. It is often an advantage to use a Company instead of an individuals effort.
I Believe that the misuse of organisational structures have a much longer history than the article suggests.
Vincento Ruggiero is a researcher who stresses the organisational aspect of Corporate crime.
A secondary concept could be economic crime.
A Company abroad could have a credit card or cash cards wich the perpetrator could use for consumption, even extensive consumption. The perpetrator could use this card for domestic or international consumption. This is a method instead of cash.
Author Response
Many thanks for the useful comments on the manuscript. Please fine the responses in the attached document.

Reviewer 2 Report
The paper provides a detailed and innovative approach towards a very interesting subject matter. The paper provides a comprehensive commentary and is well written. Reference has been made to the relevant primary and secondary sources. The research methodology is appropriate for a paper of this nature. The case studies add clear value to the theme of the paper. The conclusions are well thought out. A very interesting paper.
Author Response
Many thanks for reviewing this manuscript. Please find the response in the attached document.
